# CONTRASTIVE EXAMPLE-BASED CONTROL

## ABSTRACT

While there are many real-world problems that might benefit from reinforcement learning, these problems rarely fit into the MDP mold: interacting with the environment is often prohibitively expensive and specifying reward functions is challenging. Motivated by these challenges, prior work has developed data-driven approaches that learn entirely from samples from the transition dynamics and examples of high-return states. These methods typically learn a reward function from the high-return states, use that reward function to label the transitions, and then apply an offline RL algorithm to these transitions. While these methods can achieve good results on many tasks, they can be complex, carefully regularizing the reward function and using temporal difference updates. In this paper, we propose a simple and scalable approach to offline example-based control. Unlike prior approaches (e.g., ORIL, VICE, PURL) that learn a reward function, our method will learn an implicit model of multi-step transitions. We show that this implicit model can represent the Q-values for the example-based control problem. Thus, whereas a learned reward function must be combined with an RL algorithm to determine good actions, our model can directly be used to determine these good actions. Across a range of state-based and image-based offline control tasks, we find that our method outperforms baselines that use learned reward functions.

## 1 INTRODUCTION

Reinforcement learning is typically framed as the problem of maximizing a given reward function. However, in many real-world situations, it is more natural for users to define what they want an agent to do with examples of successful outcomes (Fu et al., 2018b; Zolna et al., 2020a; Xu & Denil, 2019; Eysenbach et al., 2021). For example, a user that wants their robot to pack laundry into a washing machine might provide multiple examples of states where the laundry has been packed correctly. This problem setting is often seen as a variant of inverse reinforcement learning (Fu et al., 2018b), where the aim is to learn only from examples of successful outcomes, rather than from expert demonstrations. To solve this problem, the agent must both figure out what constitutes task success (i.e., what the examples have in common) and how to achieve such successful outcomes.

In this paper, our aim is to address this problem setting in the case where the agent must learn from offline data. The offline setting forces the RL agent to learn without trial and error. Instead, the agent must infer the outcomes of potential actions from the provided data, while also relating these inferred outcomes to the success examples. We will refer to this problem of offline RL with success examples as *offline example-based control*.

Most prior approaches involve two steps: *first* learning a reward function, and *second* combining it with an RL method to recover a policy (Fu et al., 2018b; Zolna et al., 2020a; Xu & Denil, 2019). While such approaches can achieve excellent results when provided sufficient data (Kalashnikov et al., 2021; Zolna et al., 2020a), learning the reward function is challenging when the number of success examples is small (Li et al., 2021; Zolna et al., 2020a). These prior approaches are relatively complex (e.g., they use temporal difference learning) and have many hyperparameters.

Our aim is to provide a simple an scalable approach that avoids the challenges of reward learning. The main idea will be a learn a certain type of dynamics model. Then, using that model to predict the probabilities of reaching each of the success examples, we will be able to estimate the Q-values for every state and action. This approach does not require learning a reward function and does not use an offline RL algorithm as a subroutine. The key design decision is the type of model; we

will use an implicit model of the time-averaged future (precisely, the discounted state occupancy measure). This decision means that our model reasons across multiple time steps but will not output high-dimensional observations (only a scalar number). A limitation of this approach is that it will correspond to a single step of policy improvement: the dynamics model corresponds to the dynamics of the behavioral policy, not of the reward-maximizing policy. While this means that our method is not guaranteed to yield the optimal policy, our experiments nevertheless show that our approach outperforms multi-step RL methods.

The main contribution of this paper is an offline RL method ("Learning to Achieve Examples Offline") that learns a policy from examples of high-reward states. The key idea behind this method is an implicit dynamics model, which represents the probability of reaching states at some point in the future. We show that this model can directly estimate the the probability of reaching successful outcomes. Our method is simpler than prior approaches based on reward classifiers, and achieves better results. Our experiments demonstrate that our method can successfully solve offline RL problems from examples of high-return states on three manipulation tasks, including from image-based observations.

## 2 RELATED WORK

**Reward learning.** To overcome the challenge of hand-engineering reward functions for RL, prior methods either use supervised learning or adversarial training to learn a policy that matches the expert behavior given by the demonstration (imitation learning) (Pomerleau, 1988; Ross et al., 2011; Ho & Ermon, 2016; Spencer et al., 2021) or learn a reward function from demonstrations and optimize the policy with the learned reward through trial and error (inverse RL) (Ng & Russell, 2000; Abbeel & Ng, 2004; Ratliff et al., 2006; Ziebart et al., 2008; Finn et al., 2016; Fu et al., 2018a). However, providing full demonstrations complete with agent actions is often difficult, therefore, recent works have focused on the setting where only a set of user-specified goal states or human videos are available (Fu et al., 2018b; Singh et al., 2019; Kalashnikov et al., 2021; Xie et al., 2018; Eysenbach et al., 2021; Chen et al., 2021). These reward learning approaches have shown successes in real-world robotic manipulation tasks from high-dimensional image inputs (Finn et al., 2016; Singh et al., 2019; Zhu et al., 2020; Chen et al., 2021). Nevertheless, to combat covariate shift that could lead the policy to drift away from the expert distribution, these methods usually require significant online interaction. Unlike these works that study online settings, we consider learning visuomotor skills from offline datasets.

**Offline RL.** Offline RL (Ernst et al., 2005; Riedmiller, 2005; Lange et al., 2012; Levine et al., 2020) studies the problem of learning a policy from a static dataset without online data collection in the environment, which has shown promising results in robotic manipulation (Kalashnikov et al., 2018; Mandlekar et al., 2020; Rafailov et al., 2021; Singh et al., 2020; Julian et al., 2020; Kalashnikov et al., 2021). Prior offline RL methods focus on the challenge of policy distribution shift, by developing a variety of techniques such as regularization between the learned policy and the behavior policy of the dataset using direct policy constraints (Fujimoto et al., 2018; Liu et al., 2020; Jaques et al., 2019; Wu et al., 2019; Zhou et al., 2020; Kumar et al., 2019; Siegel et al., 2020; Peng et al., 2019; Fujimoto & Gu, 2021; Ghasemipour et al., 2021), learning conservative Q-functions (Kumar et al., 2020; Kostrikov et al., 2021; Yu et al., 2021; Sinha & Garg, 2021), and penalizing out-of-distribution states generated by learned dynamics models (Kidambi et al., 2020; Yu et al., 2020; Matsushima et al., 2020; Argenson & Dulac-Arnold, 2020; Swazinna et al., 2020; Rafailov et al., 2021; Lee et al., 2021; Yu et al., 2021).

While prior works combat overestimation caused by distribution shift, they require reward annotations of each datapoint in the large offline dataset. Practical approaches have used manual reward sketching to train a reward model (Cabi et al., 2019; Konyushkova et al., 2020; Rafailov et al., 2021) or heuristical reward functions (Yu et al., 2022). Others have considered offline learning from demonstrations, without access to a pre-defined reward function (Mandlekar et al., 2020; Zolna et al., 2020a; Xu et al., 2022; Jarboui & Perchet, 2021), however they require access to high-quality demonstrations data. In contrast, our method: *(1)* addresses distributional shift induced by both the learned policy and the reward function in a principled way, *(2)* only requires user-provided goal states and *(3)* does not require expert-quality data, resulting in an effective and practical offline reward learning scheme.

## 3 LEARNING TO ACHIEVE EXAMPLES OFFLINE

RL methods that operate in the offline setting typically require regularization, and our method will employ regularization in two ways. First, we regularize the policy with an additional behavioral cloning term, which penalizes the policy for sampling out-of-distribution actions. Second, our method uses the Q-function for the behavioral policy, so it performs one (not many) step of policy improvement. These regularizers mean that our approach is not guaranteed the yield the optimal policy.

### 3.1 PRELIMINARIES

We assume that an agent interacts with an MDP with states $s \in \mathcal{S}$, actions $a$, a state-only reward function $r(s) \geq 0$, initial state distribution $p_0(s_0)$ and dynamics $p(s_{t+1} \mid s_t, a_t)$. We use $\tau = (s_0, a_0, s_1, a_1, \cdots)$ to denote an infinite-length trajectory. The likelihood of a trajectory under a policy $\pi(a \mid s)$ is

$$\pi(\tau) = p_0(s_0) \prod_{t=0}^{\infty} p(s_{t+1} \mid s_t, a_t)\pi(a_t \mid s_t). \tag{1}$$

The objective is to learn a policy $\pi(a \mid s)$ that maximizes the expected, $\gamma$-discounted sum of rewards:

$$\max_{\pi} \mathbb{E}_{\pi(\tau)} \left[ \sum_{t=0}^{\infty} \gamma^t r(s_t) \right]. \tag{2}$$

We define the Q-function for policy $\pi$ as the expected discounted sum of returns, conditioned on an initial state and action:

$$Q^{\pi}(s, a) \triangleq \mathbb{E}_{\pi(\tau)} \left[ \sum_{t=0}^{\infty} \gamma^t r(s_t) \Big|_{a_0=a}^{s_0=s} \right]. \tag{3}$$

We will focus on the offline (i.e., batch RL) setting. Instead of learning by interacting with the environment (i.e., via trial and error), the RL agent will receive as input a dataset of trajectories $\mathcal{D}_{\tau} = \{\tau \sim \beta(\tau)\}$ collected by a behavioral policy $\beta(a \mid s)$. We will use $Q^{\beta}(s, a)$ to denote the Q-function of the behavioral policy.

**Specifying the reward function.** In many real-world applications, specifying and measure a scalar reward function is challenging, but providing examples of good states (i.e., those which would receive high rewards) is straightforward. Thus, we follow prior work (Fu et al., 2018b; Zolna et al., 2020a; Eysenbach et al., 2021; Xu & Denil, 2019; Zolna et al., 2020b) in assuming that the agent does not observe scalar rewards (i.e., $\mathcal{D}_{\tau}$ does not contain reward information). Instead, the agent receives as input a dataset $\mathcal{D}_* = \{s^*\}$ of high-reward states $s^* \in \mathcal{S}$. These high-reward states are examples of good outcomes, which the agent would like to achieve. The high-reward states are not labeled with their specific reward value.

To make the control problem well defined, we must relate these success examples to the reward function. We do this by assuming that the frequency of each success example is proportional to its reward: very good states are more likely to appear (and be duplicated) as success examples.

**Assumption 1.** *Let $p_{\tau}(s)$ be the empirical probability density of state $s$ in the trajectory dataset, and let $p_*(s)$ as the empirical probability density of state $s$ under the high-reward state dataset. We assume that there exists a positive constant $c$ such that*

$$r(s) = c\frac{p_*(s)}{p_{\tau}(s)} \quad \text{for all states } s. \tag{4}$$

This is the same assumption as Eysenbach et al. (2021), and similar assumptions are implicitly made in prior work (). This assumption is important because shows how example-based control is universal: for any reward function, we can specify the corresponding example-based problem by constructing a dataset of success examples that are sampled according to their rewards. We assumed that rewards are non-negative so that these sampling probabilities are positive.

This assumption can also be read in reverse. When a user constructs a dataset of success examples in an arbitrary fashion, they are implicitly defining a reward function. In the tabular setting, the

(implicit) reward function for state $s$ is the count of the times $s$ occurs in the dataset of success examples.

Compared with goal-conditioned RL (Kaelbling, 1993), defining tasks via success examples is more general. By identifying what all the success examples have in common (e.g., laundry is folded), the RL agent can learn what is necessary to solve the task and what is irrelevant (e.g., the color of the clothes in the laundry).

We now can define our problem statement as follows:

**Definition 1.** *In the **offline example-based control** problem, a learning algorithm receives as input a dataset of trajectories $\mathcal{D}_\tau = \{\tau\}$ and a dataset of successful outcomes $\mathcal{D}_* = \{s\}$ satisfying Assumption 1. The aim is to output a policy that maximizes the RL objective (Eq. 2).*

This problem setting is appealing because it mirrors many practical RL applications: a user has access to historical data from past experience, but collecting new experience is prohibitively expensive. Moreoever, this problem setting can mitigate the challenges of reward function design (). Rather than having to implement a reward function and add instruments to measure the corresponding components, the users need only provide a handful of observations that solved the task. This problem setting is similar to imitation learning, in the sense that the only inputs are data. However, unlike imitation learning, in this problem setting the high-reward states are not labeled with actions, and these high-reward states may not necessarily contain entire trajectories.

**An implicit model of successor representations.** Our method will estimate the discounted state occupancy measure,

$$p^\beta(s_{t+} = s \mid s_0, a_0) \triangleq (1 - \gamma) \sum_{t=0}^\infty \gamma^t p_t^\pi(s_t = s \mid s_0, a_0),$$
(5)

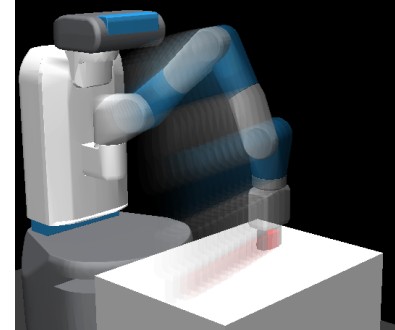

Figure 1: Our method will estimate the discounted state occupancy measure using contrastive learning. Note that this is the average over many future states.

where $p_t^\beta(s_t \mid s, a)$ is the probability of policy $\beta(a \mid s)$ visiting state $s_t$ after exactly $t$ time steps. Unlike the transition function $p(s_{t+1} \mid s_t, a_t)$, the discounted state occupancy measure indicates the probability of visiting a state at any point in the future, not just at the immediate next time step. In tabular settings, this distribution corresponds to the successor representations (Dayan, 1993). To handle continuous settings, we will use the contrastive approach from recent work (Mazoure et al., 2020; Eysenbach et al., 2022). We will learn a function $f(s, a, s_f) \in \mathbb{R}$ takes as input an initial state-action pair as well as a candidate future state, and outputs a score estimating the likelihood that $s_f$ is a real future state. The loss function is a standard contrastive learning loss(e.g., Ma & Collins (2018)), where positive examples are triplets of a state, action, and future state:

$$\max_f \mathcal{L}(f; \mathcal{D}_\tau) \triangleq \mathbb{E}_{p(s,a), s_f \sim p^\beta(s_{t+}|s,a)} [\log \sigma(f(s, a, s_f))] + \mathbb{E}_{p(s,a), s_f \sim p(s)} [\log(1 - \sigma(f(s, a, s_f)))],$$
(6)

where $\sigma(\cdot)$ is the sigmoid function. At optimality, the implicit dynamics model encodes the discounted state occupancy measure:

$$f^*(s, a, s_f) = \log p^\beta(s_{t+} = s_f \mid s, a) - \log p_\tau(s_f).$$
(7)

We visualize this implicit dynamics model in Fig. 1. Note that this dynamics model is policy dependent. Because it is trained with data collected from one policy ($\beta(a \mid s)$), it will correspond to the probability that *that* policy visits states in the future. Because of this, our method will result in estimating the value function for the behavioral policy (akin to 1-step RL (Brandfonbrener et al., 2021)), and will not perform multiple steps of policy improvement.

Intuitively, the training of this implicit model resembles hindsight relabeling (Kaelbling, 1993; Andrychowicz et al., 2017). However, it is generally unclear how to use hindsight relabeling for single-task problems. Despite being a single-task method, our method will be able to make use of hindsight relabeling to train the dynamics model.

## 3.2 Deriving Our Method

The key idea behind out method is that this implicit dynamics model can be used to represent the Q-values for the example-based problem, up to a constant.

**Lemma 3.1.** *Assume that the implicit dynamics model is learned without errors. Then the Q-function for the data collection policy $\beta(a \mid s)$ can be expressed in terms of this implicit dynamics model:*

$$Q^\beta(s,a) = \frac{c}{1-\gamma}\mathbb{E}_{p_*(s^*)}\left[e^{f(s,a,s^*)}\right]. \tag{8}$$

The proof follows by substituting Assumption 1 into the definition of Q-values (Eq. 3):

*Proof.*

$$Q^\beta(s,a) = \mathbb{E}_{\beta(\tau)}\left[\sum_{t=0}^{\infty}\gamma^t r(s_t)\Big|_{a_0=a}^{s_0=s}\right] = \frac{1}{1-\gamma}\int p^\beta(s_{t+}=s^* \mid s,a)r(s^*)ds^*$$

$$= \frac{1}{1-\gamma}\int p^\beta(s_{t+}=s^* \mid s,a)c\frac{p_*(s^*)}{p_\tau(s^*)}ds^*$$

$$= \frac{c}{1-\gamma}\int p_*(s^*)e^{f(s,a,s^*)}ds^* = \frac{c}{1-\gamma}\mathbb{E}_{s^*\sim p_*(s)}\left[e^{f(s,a,s^*)}\right].$$

$\square$

So, after learning the implicit dynamics model, we can estimate the Q-values by averaging this model's predictions across the success examples. We will update the policy using Q-values estimated in this manner, plus a regularization term:

$$\min_\pi \mathcal{L}(\pi; f, \mathcal{D}_*) \triangleq -(1-\lambda)\mathbb{E}_{\pi(a|s)p(s),s^*\sim\mathcal{D}_*}\left[e^{f(s,a,s^*)}\right] - \lambda\mathbb{E}_{s,a\sim\mathcal{D}_\tau}\left[\log\pi(a \mid s)\right]. \tag{9}$$

In our experiments, we use a weak regularization coefficient of $\lambda = 0.05$.

It is worth comparing this approach to prior methods based on learned reward functions (Xu & Denil, 2019; Fu et al., 2018b; Zolna et al., 2020a). Those methods learn a reward function from the success examples, and use that learned reward function to synthetically label the dataset of trajectories. Both approaches can be interpreted as learning a function on one of the datasets and then applying that function to the other dataset. Because it is easier to fit a function when given large quantities of data, we predict that our approach will outperform the learned reward function approach when the number of success examples is small, relative to the number of unlabeled trajectories. Other prior methods (Eysenbach et al., 2021; Reddy et al., 2020) avoid learning reward functions by proposing TD update rules that are applied to both the unlabeled transitions and the high-return states. However, because these methods have yet to be adapted to the offline RL setting, we will focus our comparisons on the reward-learning methods.

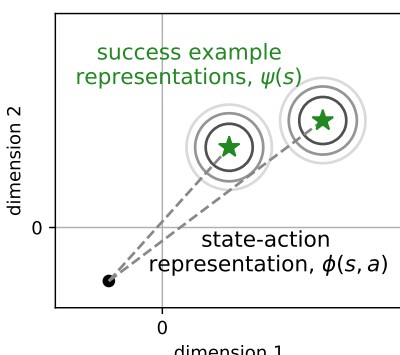

Figure 2: If the state-action representation $\phi(s,a)$ is close to the representation of a high-return state $\psi(s)$, then the policy is likely to visit that state. Our method estimates Q-values by combining the distances to all the high-return states (Eq. 3).

## 3.3 A Geometric Perspective

Before presenting the complete RL algorithm, we provide a geometric perspective on the representations learned by our method. Our implicit models learns a representation of state-action pairs $\phi(s,a)$ as well as a representation of future states $\psi(s)$. One way that our method can optimize these representations is by treating $\phi(s,a)$ as a prediction for the future representations.[1] Each

---

[1]Our method can also learn the opposite, where $\psi(s)$ is a prediction for the previous representations.

of the high-return states can be mapped to the same representation space. To determine whether a state-action pair has a large or small Q-value, we can simply see whether the predicted representation $\phi(s, a)$ is close to the representations of any of the success examples. Our method learns these representations so that the Q-values are directly related to the Euclidean distances[2] from each success example. Thus, our method can be interpreted as learning a representation space such that estimating Q-values corresponds to simple geometric operations (kernel smoothing with an RBF kernel (Hastie et al., 2009, Chpt. 6)) on the learned representations. While the example-based control problem is more general than goal-conditioned RL (see Sec. 3.1), we can recover goal-conditioned RL as a special case by using a single success example.

### 3.4 A Complete Algorithm

We now build a complete offline RL algorithm based on these Q-functions. We will call our method "Learning to Achieve Examples Offline," or LAEO for short. Our algorithm will resemble one-step RL methods, but differ in how the Q-function is trained. After learning the implicit dynamics model (and, hence, Q-function) we will optimize the policy. The objective for the policy is maximizing Q-values plus a behavioral cloning regularization term, which penalizes sampling unseen actions:

$$\max_{\pi} (1 - \lambda)\mathbb{E}_{\pi(a|s)p_{\tau}(s)}\left[Q(s, a)\right] + \lambda\mathbb{E}_{(s,a)\sim p_{\tau}(s,a)}\left[\log \pi(a \mid s)\right]$$

$$= (1 - \lambda)\mathbb{E}_{\pi(a|s),s^*\sim p_*(s)}\left[e^{f(s,a,s^*)}\right] + \lambda\mathbb{E}_{(s,a)\sim p_{\tau}(s,a)}\left[\log \pi(a \mid s)\right]. \tag{10}$$

As noted above, this is a one-step RL method: it updates the policy to maximize the Q-values of the behavioral policy. Performing just a single step of policy improvement can be viewed as a form of regularization in RL, in the same spirit as early stopping is a form of regularization in supervised learning. Prior work has found that one-step RL methods can

---

**Algorithm 1** Learning to Achieve Examples Offline

1: **Inputs**: dataset of trajectories $\mathcal{D} = \{\tau\}$, dataset of high-return states $\mathcal{D}_* = \{s\}$.
2: Learn the model via contrastive learning: $f \leftarrow \arg\min_f \mathcal{L}(f; \mathcal{D}_{\tau})$ ▷ Eq. 9
3: Learn the policy: $\pi \leftarrow \arg\min_{\pi} \mathcal{L}(\pi; f, \mathcal{D}_*)$ ▷ Eq. 10
4: **return** policy $\pi(a \mid s)$

---

perform well in the offline RL setting. Because our method performs only a single step of policy improvement, we are not guaranteed that it will converge to the reward-maximizing policy. We summarize the complete algorithm in Alg. 1.

## 4 Experiments

Our experiments will test whether LAEO can effectively solve offline RL tasks that are specified by examples of high-return states, rather than via scalar reward functions. Our experiments will study when our approach outperforms prior approaches based on learned reward functions. We will look not just at the performance relative to baselines on state-based and image-based tasks, but also how that performance depends on the input datasets. Additional experiments study how the example-based problem setting allows agents to learn a generalized notion of each task, and study whether our method can solve partially observed tasks. We include full hyperparameters and implementation details in Appendix A. Code will be released.

**Baselines.** Our main point of comparison will be prior methods that use learned reward functions: PURL (Xu & Denil, 2019), and ORIL (Zolna et al., 2020a). The main difference between these methods is the loss function used to train reward function: ORIL uses binary cross entropy loss while PURL uses a positive-unlabeled loss (Xu & Denil, 2019). After learning the reward function, each of these methods applies an off-the-shelf RL algorithm. We will implement all baselines using the TD3+BC (Fujimoto & Gu, 2021) offline RL algorithm. These offline RL methods achieve good performance on tasks specified via reward functions (Kostrikov et al., 2021; Brandfonbrener et al., 2021; Fujimoto & Gu, 2021).

---

[2]When representations are normalized, the dot product is equivalent to the Euclidean norm. We find that unnormalized features work better in our experiments.

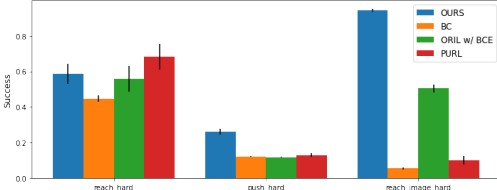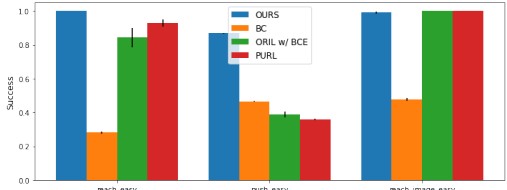

Figure 3: **Offline RL from Examples**: LAEO outperforms prior offline RL methods, including those that learn a separate reward function (ORIL, PURL). The gap is especially large on the image-based tasks (right).

Figure 4: **Effect of dataset size**: *(Left)* LAEO outperforms PURL and ORIL when only a small number of high-return states are given, likely because ORIL and PURL struggle to fit the learned reward function on such a small dataset. *(Right)* LAEO continues to improve when trained with more reward-free trajectories, while the ORIL's and PURL's performances plateau.

**Benchmark comparison.** We start by comparing the performance of LAEO to these baselines on four manipulation tasks. `FetchReach-state` and `FetchPush-state` are two manipulation tasks from Plappert et al. (2018) that use state-based observations. `FetchReach-image` and `FetchPush-image` are the same tasks but with image-based observations. For each of these tasks, we collect a dataset of medium quality by training an RL agent on the ground-truth reward function and collecting data using an intermediate training checkpoint. We will also include an ablation experiment where we use high-quality data, collected from the final checkpoint of this oracle policy. Since we are in the offline setting, we do no report results as a function of number of environment interactions (which is always 0). Instead, we will report results after 7000 training epochs, each of which corresponds to one pass over the offlien dataset

We report results in Fig. 3. We observe that all methods (LAEO, PURL, ORIL) perform similarly on `FetchReach` and `FetchReach-image`. This is likely because these are relatively easy tasks, and each of these methods is able to achieve an almost perfect success rate. On `FetchPush`, LAEO outperforms all of the baselines by a significant margin. Recall that the main difference between LAEO and PURL/ORIL is by learning a dynamics model, rather than the reward function. These experiments suggest that for tasks with more complex dynamics, such as `FetchPush`, learning a dynamics model can achieve better performance than is achieved by model-free reward learning methods.

**Varying the input data.** One important criterion for offline RL algorithms is that they can learn from datasets of varying quality, varying from expert-level demonstrations to low-quality data collected from a near-random policy. Our next experiment studies how the dataset composition affects LAEO and the baselines. We will use three tasks: `FetchReach`, `FetchPush`, and `FetchReach-image`. On each of these tasks, we generate a high-quality dataset and a low quality data. These datasets are generated by training an oracle policy using the ground-truth reward function, and using a partially-trained policy (for the low-quality dataset) and the fully-trained policy (for the high-quality dataset). The high quality datasets have between $45\% - 5 - \%$ success rates, and the low quality datasets have between $8\% - 12\%$ success rates.

We report results in Fig. 4. We observe that on `FetchReach-image` (hard) our method significantly outperforms the baselines. This suggests that in image based domains with low quality datasets, learning a dynamics model instead of a reward function is especially beneficial.

## 5 CONCLUSION

In this paper, we presented an RL algorithm aimed at settings where data collection and reward specification are difficult. Our method learns from a combination of high-return states and reward-free trajectories, integrating these two types of information to learn reward-maximizing policies. Whereas prior methods perform this integration by learning a reward function and then applying an off-the-shelf RL algorithm, ours learns an implicit dynamics model. Not only is our method simpler (not additional RL algorithm required!), but also it achieves higher success rates on the benchmark

tasks. Because our reinforcement learning method resembles prior representation learning methods (Mazoure et al., 2020; Nair et al., 2022), we believe that scaling this method to very large offline datasets is an important direction for future work.

**Limitations.** One limitation of our method is that it corresponds to a single step of policy improvement. Extending the method to be less regularized by performing multiple steps of policy improvement remains an open problem. Second, our method learns to solve a single task, whereas we arguably would like RL agents that could solve multiple tasks. Extending LAEO to the multi-task setting is an exciting direction for future work.

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

## A  EXPERIMENTAL DETAILS

We implemented our method and all baselines using the ACME framework (Hoffman et al., 2020).

- Batch size: 1024 for state based experiments, 256 for image based experiments
- Training iterations: $448,000$
- Representation dimension: 256
- Reward learning loss (for baselines): binary cross entropy (for ORIL) and positive unlabeled (for PURL)
- Critic architecture: Two-layer MLP with hidden sizes of 1024. ReLU activations used between layers.

- Reward function architecture (for baselines): Two-layer MLP with hidden sizes of 1024. ReLU activations used between layers.
- Actor learning rate: $3 \times 10^{-4}$
- Critic learning rate: $3 \times 10^{-4}$
- Reward learning rate (for baselines): $1 \times 10^{-4}$
- $\eta$ for PU loss: $0.5$
- Size of offline datasets: Each dataset consists of $4000$ trajectories of length $50$, for $200,000$ total transitions.

