# OpenReview forum: "Contrastive Example-Based Control"
_NeurIPS.cc/2022/Workshop/Offline_RL — Offline RL Workshop NeurIPS 2022_

### Official Review · Reviewer_6hsa · 2022-10-07
**Interesting perspective, but questions on the necessity of example-based control**

**Rating:** 6
**Confidence:** 5

**Review:**

This work shows that example based control could be applied to Offline RL setting, and shows promising results on some control datasets.

Questions:
0. How were the high-return demonstrations collected?

1. How does the results of this work compare to "Contrastive Learning as Goal-Conditioned Reinforcement Learning", which also demonstrated that their method could be applied offline? https://arxiv.org/pdf/2206.07568.pdf
2. Assumption 1 shows that "When a user constructs a dataset of success examples in an arbitrary fashion, they are implicitly defining a reward function". However, since we're in an offline setting, can't we just label the good states +1 reward and bad states 0 reward and ask the decision transformer to do something similar (attribute value to rollouts)?
3. Experiments seem to show good performance on datasets where we already have reward, and examples seem to have been generated from reward function. However, isn't the goal to work without reward? Suppose we don't have access to the true reward, how do we generate examples? Are the examples generated as good as those generated with reward function?
4. Would training an ensemble be computationally inefficient for harder tasks?
5. Have you tried other d4rl benchmarks?

Notes:
Page 3: " similar assumptions are implicitly made in prior work ()" Citation broken.

---

### Official Review · Reviewer_vdvL · 2022-10-19
**Valuable contribution, requires major improvements in terms of presentation**

**Rating:** 7
**Confidence:** 4

**Review:**

This work introduces an offline RL algorithm, Learning to Achieve examples Offline (LAEO), that learns a policy from examples of high-reward states. The authors propose to learn an implicit dynamics model which encodes the state occupancy measure. Then using the learned model, the policy is optimized. The problem of learning from a few examples of successful outcomes is well-motivated and important. In general, I found the idea of the paper interesting and valuable to the community. However, the presentation of the paper requires major improvements:

- A few citations are missing. For example 2nd line below Assumption 1 and 3rd line after Definition 1.
- Figure 3 is missing! and Figure 4's labels are hard to read.
- In terms of writing, there are several sentences either hard to understand or not complete. e.g in the abstract: "While these methods can
achieve good results on many tasks, they can be complex, carefully regularizing the reward function and using temporal difference updates."

The paper can also be improved by adding more explanations on:
- Why is Equation 7 correct? Mathematically or intuitively.
- For assumption 1, prior work is referenced, but it might be useful to add a few sentences about why the chosen reward function makes sense.